# Collateral impacts of pandemic COVID-19 drive the nosocomial spread of antibiotic resistance: A modelling study

**David R. M. Smith**[1,2,3,4]*, **George Shirreff**[1,2,3], **Laura Temime**[3,5ʘ], **Lulla Opatowski**[1,2ʘ]

1 Institut Pasteur, Université Paris Cité, Epidemiology and Modelling of Antibiotic Evasion (EMAE), Paris, France, 2 Université Paris-Saclay, UVSQ, Inserm, CESP, Anti-infective evasion and pharmacoepidemiology team, Montigny-Le-Bretonneux, France, 3 Modélisation, épidémiologie et surveillance des risques sanitaires (MESuRS), Conservatoire national des arts et métiers, Paris, France, 4 Health Economics Research Centre, Nuffield Department of Population Health, University of Oxford, Oxford, United Kingdom, 5 PACRI unit, Institut Pasteur, Conservatoire national des arts et métiers, Paris, France

ʘ These authors contributed equally to this work.
* david.smith@ndph.ox.ac.uk

## Abstract

### Background

Circulation of multidrug-resistant bacteria (MRB) in healthcare facilities is a major public health problem. These settings have been greatly impacted by the Coronavirus Disease 2019 (COVID-19) pandemic, notably due to surges in COVID-19 caseloads and the implementation of infection control measures. We sought to evaluate how such collateral impacts of COVID-19 impacted the nosocomial spread of MRB in an early pandemic context.

### Methods and findings

We developed a mathematical model in which Severe Acute Respiratory Syndrome Coronavirus 2 (SARS-CoV-2) and MRB cocirculate among patients and staff in a theoretical hospital population. Responses to COVID-19 were captured mechanistically via a range of parameters that reflect impacts of SARS-CoV-2 outbreaks on factors relevant for pathogen transmission. COVID-19 responses include both "policy responses" willingly enacted to limit SARS-CoV-2 transmission (e.g., universal masking, patient lockdown, and reinforced hand hygiene) and "caseload responses" unwillingly resulting from surges in COVID-19 caseloads (e.g., abandonment of antibiotic stewardship, disorganization of infection control programmes, and extended length of stay for COVID-19 patients). We conducted 2 main sets of model simulations, in which we quantified impacts of SARS-CoV-2 outbreaks on MRB colonization incidence and antibiotic resistance rates (the share of colonization due to antibiotic-resistant versus antibiotic-sensitive strains).

The first set of simulations represents diverse MRB and nosocomial environments, accounting for high levels of heterogeneity across bacterial parameters (e.g., rates of transmission, antibiotic sensitivity, and colonization prevalence among newly admitted patients) and hospital parameters (e.g., rates of interindividual contact, antibiotic exposure, and patient admission/discharge). On average, COVID-19 control policies coincided with MRB

**Data Availability Statement:** All data used in and produced by this study are available at https://github.com/drmsmith/covR/.

**Funding:** The Epidemiology & Modelling of Antibiotic Evasion team and the Anti-infective Evasion and Pharmacoepidemiology team received

funding from the MODCOV project from the Fondation de France as part of the alliance framework "Tous unis contre le virus" (#106059), the Université Paris-Saclay (AAP Covid-19 2020) and the French National Research Agency and the "Investissement d'Avenir" program, Laboratoire d'Excellence "Integrative Biology of Emerging Infectious Diseases" (ANR-10-LABX-62- IBEID). Researchers were also supported by research grants from the French National Research Agency (SPHINX-17-CE36-0008-01 to L.T and L.O) and the Canadian Institutes of Health Research (Doctoral Foreign Study Award #164263 to D.R.M.S.). The funders had no role in study design, data collection and analysis, decision to publish, or preparation of the manuscript.

**Competing interests:** I have read the journal's policy and the authors of this manuscript have the following competing interests: L.O. reports grants from Pfizer outside the submitted work. Authors declare no other competing interests

**Abbreviations:** COVID-19, Coronavirus Disease 2019; ESBL, extended-spectrum beta-lactamase-producing; HCW, healthcare worker; IPC, infection prevention and control; MRB, multidrug-resistant bacteria; MRSA, methicillin-resistant *Staphylococcus aureus*; ODE, ordinary differential equation; SARS-CoV-2, Severe Acute Respiratory Syndrome Coronavirus 2; SEIR, Susceptible-Exposed-Infectious-Recovered; UI, uncertainty interval; VRE, vancomycin-resistant Enterococci.

prevention, including 28.2% [95% uncertainty interval: 2.5%, 60.2%] fewer incident cases of patient MRB colonization. Conversely, surges in COVID-19 caseloads favoured MRB transmission, resulting in a 13.8% [−3.5%, 77.0%] increase in colonization incidence and a 10.4% [0.2%, 46.9%] increase in antibiotic resistance rates in the absence of concomitant COVID-19 control policies. When COVID-19 policy responses and caseload responses were combined, MRB colonization incidence decreased by 24.2% [−7.8%, 59.3%], while resistance rates increased by 2.9% [−5.4%, 23.2%]. Impacts of COVID-19 responses varied across patients and staff and their respective routes of pathogen acquisition.

The second set of simulations was tailored to specific hospital wards and nosocomial bacteria (methicillin-resistant *Staphylococcus aureus*, extended-spectrum beta-lactamase producing *Escherichia coli*). Consequences of nosocomial SARS-CoV-2 outbreaks were found to be highly context specific, with impacts depending on the specific ward and bacteria evaluated. In particular, SARS-CoV-2 outbreaks significantly impacted patient MRB colonization only in settings with high underlying risk of bacterial transmission. Yet across settings and species, antibiotic resistance burden was reduced in facilities with timelier implementation of effective COVID-19 control policies.

## Conclusions

Our model suggests that surges in nosocomial SARS-CoV-2 transmission generate selection for the spread of antibiotic-resistant bacteria. Timely implementation of efficient COVID-19 control measures thus has 2-fold benefits, preventing the transmission of both SARS-CoV-2 and MRB, and highlighting antibiotic resistance control as a collateral benefit of pandemic preparedness.

## Author summary

### Why was this study done?

- Antibiotic resistance is a major global health problem, and healthcare settings are hot-spots for the spread of antibiotic-resistant bacteria.

- Healthcare settings have been heavily impacted by the Coronavirus Disease 2019 (COVID-19) pandemic, in particular due to sudden surges of COVID-19 cases, the ensuing disorganization of care delivery, and the enactment of infection control measures designed to curb viral transmission.

- The COVID-19 pandemic has led to shifts in the epidemiological dynamics of diverse infectious diseases, but its impacts on the spread of antibiotic-resistant bacteria remain poorly understood, due in part to the largely unobserved nature of bacterial colonization.

### What did the researchers do and find?

- A mathematical model was developed and used to assess how outbreaks of Severe Acute Respiratory Syndrome Coronavirus 2 (SARS-CoV-2) in healthcare settings may impact patient colonization with antibiotic-resistant bacteria.

- Surges in COVID-19 cases fostered conditions favourable for bacterial transmission, on average resulting in a 14% increase in colonization acquisition and a 10% increase in rates of antibiotic resistance.

- Conversely, the implementation of COVID-19 control measures provided the unintended benefit of limiting bacterial spread, leading to a 28% reduction in patient acquisition of drug-resistant bacteria.

- Impacts of SARS-CoV-2 outbreaks on antibiotic resistance were found to depend fundamentally on the particular characteristics of different hospital wards and bacterial species, but more timely implementation of effective COVID-19 control policies helped to limit the spread of antibiotic resistance across a wide range of contexts.

### What do these findings mean?

- Outbreaks of respiratory pathogens like SARS-CoV-2 risk aggravating the concomitant spread of antibiotic-resistant bacteria.

- Healthcare facilities with greater underlying risk of bacterial transmission are likely more vulnerable to surges in antibiotic resistance in the event of a pandemic.

- Limiting the spread of antibiotic resistance should be considered as a collateral benefit of pandemic preparedness initiatives that enable more efficient public health responses to counter emerging infectious threats.

## Introduction

The Coronavirus Disease 2019 (COVID-19) pandemic has impacted the epidemiology of diverse infectious diseases, including sexually transmitted infections (e.g., HIV) [1], vector-borne illnesses (e.g., dengue virus) [2], and invasive bacterial diseases (e.g., *Streptococcus pneumoniae*) [3]. Antibiotic resistance is a leading global driver of infectious morbidity and mortality [4], yet impacts of the pandemic on the transmission and control of antibiotic-resistant bacteria remain poorly understood. There are many ways by which the COVID-19 pandemic is believed to have influenced antibiotic resistance dynamics, particularly in healthcare settings, which face a disproportionately large share of the epidemiological burden of both antibiotic resistance and COVID-19. On one hand, surges in COVID-19 cases have led to conditions favourable for the proliferation of antibiotic-resistant bacteria, including hospital disorganization, increased demand on healthcare workers (HCWs), abandonment of antimicrobial stewardship programmes, and high rates of antibiotic prescribing among COVID-19 patients. On the other, public health interventions implemented to control nosocomial Severe Acute Respiratory Syndrome Coronavirus 2 (SARS-CoV-2) transmission—including patient lockdowns, hand hygiene education, and provisioning of alcohol-based hand rub—may provide the unintended benefit of preventing bacterial transmission.

Early in the pandemic, researchers and public health officials warned that COVID-19 may impact global efforts to curb antibiotic resistance [5,6]. However, epidemiological surveillance has been greatly challenged by COVID-19 [7], and studies to date report heterogeneous impacts of the pandemic on antibiotic-resistant bacteria. One review highlights decreased incidence of healthcare-associated infections caused by vancomycin-resistant Enterococci (VRE) and methicillin-resistant *Staphylococcus aureus* (MRSA) relative to pre-pandemic levels [8].

Yet in an analysis of microbiological data from 81 hospitals in the United States of America, infections due to MRSA, VRE, and multidrug-resistant gram-negative bacteria all spiked during local surges in COVID-19 cases [9]. In a United Kingdom hospital network, bloodstream infection due to MRSA and coagulase-negative staphylococci also spiked during surges in COVID-19 cases, while those due to Enterobacterales reached historic lows in 2 hospitals [10,11].

These conflicting reports suggest that impacts of COVID-19 on antibiotic resistance likely depend on the particular population, setting, and bacteria in question and may be highly context specific. Several international studies have now reported on rates of healthcare-associated infection during the pandemic [12,13], but few have reported data on bacterial colonization or transmission, on rates of antibiotic resistance among colonized patients, nor on the putative mechanisms driving potential pandemic-related shits in antibiotic resistance epidemiology [14]. Mathematical modelling is a useful tool to help disentangle the mechanisms linking the transmission dynamics of co-occurring pathogens, especially when data are limited. However, recent work suggests that models describing impacts of COVID-19 on antibiotic resistance dynamics remain scarce [15].

To anticipate and mitigate collateral impacts of SARS-CoV-2 outbreaks—and potential outbreaks of other, as-yet unknown pathogens—there is a need to better understand how the COVID-19 pandemic has both selected for and controlled against antibiotic resistance. Here, we propose a mathematical model describing the transmission of SARS-CoV-2 and commensal bacteria among patients and staff in a healthcare setting. We include mechanistic impacts of SARS-CoV-2 outbreaks on antibiotic consumption, interindividual contact behaviour, infection prevention and control (IPC) practices, and the size and make-up of the hospital population. Simulations are used to understand and quantify how outbreaks of SARS-CoV-2 may have influenced antibiotic resistance epidemiology in an early pandemic context.

## Methods

### A nosocomial transmission model for SARS-CoV-2 and antibiotic-resistant bacteria

We used ordinary differential equations (ODEs) to formalize a deterministic, compartmental model describing the transmission dynamics of SARS-CoV-2 ($V$) and a commensal bacterium ($B$) among inpatients (*pat*) admitted to a healthcare facility, and among HCWs (*hcw*) providing care to patients (**Fig 1**). SARS-CoV-2 is conceptualized as transmitting via exhaled respiratory droplets/aerosols, while bacteria are conceptualized as transmitting via fomites and physical touch. We assume no within-host ecological interactions between $V$ and $B$: bacterial colonization does not directly impact SARS-CoV-2 infection, nor does infection directly impact colonization.

SARS-CoV-2 infection is characterized by a modified Susceptible-Exposed-Infectious-Recovered (*SEIR*) process among patients and HCWs, with potential sick leave among symptomatic HCWs. The bacterium is characterized by ecological competition between antibiotic-sensitive strains ($B^S$) and antibiotic-resistant strains ($B^R$). Among patients, we consider exclusive asymptomatic bacterial colonization ($C^S$ or $C^R$), which is potentially cleared naturally (after $1/\gamma$ days) or as a result of antibiotic exposure (after $1/\sigma$ days). $B^R$ is assumed to resist a greater share of antibiotics than $B^S$, but not necessarily all antibiotics ($0 \leq r_{B^S} \leq r_{B^R} \leq 1$), and bears a fitness cost resulting in faster natural clearance ($\gamma_{B^R} \geq \gamma_{B^S}$). Among HCWs, we consider exclusive transient bacterial carriage ($T^S$ or $T^R$), which is potentially cleared via HCW decontamination ($\omega$) and depends upon HCW compliance with hand hygiene ($H$) subsequent to HCW–patient contact ($\kappa^{hcw \rightarrow pat}$). HCWs are thus conceptualized as potential vectors for

## a. hospital population and behaviour

## b. SARS-CoV-2 infection

## c. bacterial acquisition and clearance

**Fig 1. Model schematic describing the 3 levels of complexity included in the hospital population.** (**a**) Behaviours within the healthcare facility, including asymmetric contact patterns among and between patients and HCWs, patient admission and discharge, patient exposure to antibiotics, and HCW compliance with hand hygiene. (**b**) SARS-CoV-2 infection progression among patients and HCWs, modelled as a modified SEIR process. For simplicity, death is not explicitly considered. (**c**) Bacterial acquisition and clearance, including patient colonization and clearance via antibiotics, and HCW carriage and clearance via hand hygiene. HCW, healthcare worker; SARS-CoV-2, Severe Acute Respiratory Syndrome Coronavirus 2; SEIR, Susceptible-Exposed-Infectious-Recovered.

bacterial transmission (HCW colonization is not considered). With this model, antibiotics select for resistant bacteria, both at the between-host level due to preferential clearance of sensitive bacteria and at the within-host level due to increased rates of endogenous acquisition during antibiotic exposure [16].

See **S1 Appendix** section 1.1 for full model description and equations. The complete model is programmed in R, and all code is freely available at https://github.com/drmsmith/covR.

## COVID-19 responses: Policy versus caseload

Ten *COVID-19 response parameters* ($\tau_i$) were included in ODEs by changing how individuals flow through the model or modifying relevant parameter values (detailed in **Table 1**). Each parameter reflects a distinct way in which SARS-CoV-2 outbreaks impact the organization of healthcare settings and delivery of care. These parameters are normalized such that $0 \leq \tau_i \leq 1$

**Table 1. Responses to COVID-19 included in the transmission model.** See Methods for description of how COVID-19 response parameters are implemented in model equations. Rows describing policy responses are shaded blue, and rows describing caseload responses are shaded grey. Symptomatic refers to COVID-19 symptoms among individuals infected with SARS-CoV-2.

| COVID-19 response | | Evidence | Model implementation | Category/ Cause | Interpretation | |
|---|---|---|---|---|---|---|
| | | | | | $\tau = 0$ | $\tau = 1$ |
| $\tau_{as}$ | Abandoned stewardship | Reduction in antibiotic stewardship activities [17] | Increased proportion of patients exposed to antibiotics ($A$) | Antibiotics/ Caseload | No change in antibiotic use | Large increase in antibiotic use during COVID-19 surges |
| $\tau_{pl}$ | COVID-19 prescribing | COVID-19 patients receive high rates of antibiotic prescription [18] | Increased proportion of symptomatic COVID-19 patients exposed to antibiotics ($A_I$) | Antibiotics/ Policy | No excess antibiotic prescribing among symptomatic patients | All symptomatic patients receive antibiotics |
| $\tau_{cd}$ | Care disorganization | Compromised ability of HCWs to adhere to IPC best practices (e.g., due to increased workload, PPE shortages) [14] | Increased daily rate of at-risk patient–HCW contact ($\kappa^{pat \rightarrow hcw}$) | Contact/ Caseload | No change in contact behaviour | Large increase in at-risk patient–HCW contact during COVID-19 surges |
| $\tau_{pl}$ | Patient lockdown | Social interactions among patients limited or forbidden [19] | Decreased daily rate of patient–patient contact ($\kappa^{pat \rightarrow pat}$) | Contact/ Policy | No change in contact behaviour | Elimination of all patient–patient contact |
| $\tau_{um}$ | Universal masking | HCWs and patients wear face masks to prevent transmission [20] | Decreased SARS-CoV-2 transmissibility per contact ($\pi_V$) | IPC/Policy | No change in SARS-CoV-2 transmissibility | SARS-CoV-2 rendered nontransmissible (perfect mask effectiveness) |
| $\tau_{hh}$ | Hand hygiene | Increase in HCW handwashing performance [21] | Increased hand hygiene compliance ($H$) | IPC/Policy | No change in hand hygiene compliance | Perfect hand hygiene compliance |
| $\tau_{cs}$ | COVID-19 stays | COVID-19 patients remain in healthcare facility until recovered [22] | Decreased discharge rate for symptomatic COVID-19 patients ($\mu_I$) | Disease/ Caseload | No impact of SARS-CoV-2 infection on patient length of stay | All patients remain in hospital while symptomatic |
| $\tau_{ss}$ | Staff sick leave | HCWs with COVID-19 stay home from work [23] | A proportion of symptomatic HCWs removed from population for 7 days (until recovered) | Disease/ Caseload | No symptomatic staff go on sick leave | All symptomatic staff go on sick leave after being infectious for 1 day |
| $\tau_{ra}$ | Reduced admission | Decreased number of hospital admissions during COVID-19 surges [24] | Decreased patient admission rate ($\mu$) | Admission/ Caseload | No change in patient admissions | Large reduction in patient admissions during COVID-19 surges |
| $\tau_{sc}$ | Sicker casemix | Elective admissions delayed or cancelled during COVID-19 surges, restricting admissions to more critically ill patients [10] | Increased rate of antibiotic-resistant bacterial carriage among patient admissions ($f_{C^R}$) | Admission/ Caseload | No change in the probability of colonization upon admission | Large increase in the probability of colonization with resistant bacteria upon admission during COVID-19 surges |

COVID-19, Coronavirus Disease 2019; HCW, healthcare worker; IPC, infection prevention and control; PPE, personal protective equipment; SARS-CoV-2, Severe Acute Respiratory Syndrome Coronavirus 2.

(where $\tau_i = 0$ signifies no impact of COVID-19 response $i$ and $\tau_i = 1$ signifies the maximum impact of that response). Each COVID-19 response is further classified as either a *policy response* willingly enacted during SARS-CoV-2 outbreaks to limit viral transmission, or as a *caseload response* that unwillingly results from surges in COVID-19 patients or HCW infection (see **Methods** and **Figure A in S1 Appendix**). The model is structured such that these parameters have mechanistic impacts on pathogen transmission (**Figure B in S1 Appendix**).

*Policy responses* are implemented at a time $t_{policy}$ and reflect evolution of public health policy or practice within the healthcare facility over the course of the epidemic. For each of these responses ($\tau_{cp}, \tau_{pl}, \tau_{um}, \tau_{hh}$), we assume a phase-in period of duration $t_{impl}$, during which the policy is gradually adopted, such that it is implemented with full impact at time $t_{policy}+t_{impl}$. Hence, for each policy response $\tau_t$, the value over time $T_t(t, \tau_t)$ is taken as

$$T_t(t, \tau_t) = \begin{cases} 0, t < t_{policy} \\ \tau_t \times \dfrac{t - t_{policy}}{t_{impl}}, \quad t - t_{policy} < t_{impl} \\ \tau_t, t \geq t_{impl} \end{cases}$$

where

$$t_{impl} > 0.$$

Second, *caseload responses* depend on patient and/or staff SARS-CoV-2 infection prevalence, reflecting impacts of increasing COVID-19 caseloads on provisioning of care. COVID-19 stays ($\tau_{cs}$) and staff sick leave ($\tau_{ss}$) impact the numbers of infectious patients and staff in the healthcare facility, while all other dynamic caseload responses ($\tau_{as}, \tau_{cd}, \tau_{ra}, \tau_{sc}$) scale dynamically with patient infection prevalence. The quantile of the cumulative Beta distribution $B_{x(t)}(\alpha, \beta)$ corresponding with patient SARS-CoV-2 infection prevalence $x(t)$ is used, fixing $\alpha = 2$ and scaling $\beta$ by the dynamic caseload response $\tau_x$. This gives a modified time- and prevalence-dependent value $T_x(t)$,

$$T_x(t, \tau_x) = \begin{cases} 0, \tau_x = 0 \\ B_{x(t)}\left(\alpha = 2, \beta = \dfrac{1}{\tau_x}\right), \quad \tau_x > 0 \end{cases}$$

where

$$x(t) = \frac{\sum_{g \in h} I_g^{pat}(t)}{N^{pat}(t)}$$

given bacterial colonization status $g$ in a set of statuses $h$ (see **S1 Appendix** **section 1.1**). The relationship between $\tau_x$, $x(t)$, and $T_x(t)$ is visualized in **Figure A in S1 Appendix**.

## Simulations

Model simulations were conducted to quantify impacts of SARS-CoV-2 outbreaks and corresponding COVID-19 responses on the epidemiological dynamics of antibiotic resistance. Simulations aimed at representing poorly anticipated nosocomial SARS-CoV-2 outbreaks in naïve hospital populations, as in the first wave of the COVID-19 pandemic. Dynamics were simulated by solving ODEs through numerical integration using the R package *deSolve* [25]. For each simulation, endemic equilibria for bacterial carriage and colonization were found. Then, 2 cases of SARS-CoV-2 (1 patient, 1 HCW) were introduced into the facility, assuming

complete susceptibility to infection among all other individuals. Dynamics were run for a period of 180 days from SARS-CoV-2 introduction, using estimates of SARS-CoV-2 transmissibility from early 2020 from a long-term care hospital in Paris, France (see parameter values in **Table A in S1 Appendix**) [26]. In simulations tailored to this particular hospital, when all COVID-19 responses were combined with random magnitude, nosocomial SARS-CoV-2 outbreaks had complex impacts on hospital demography and healthcare-associated behaviours (**Fig 2A-2E**), with heterogeneous consequences for epidemiological dynamics of both SARS-CoV-2 and antibiotic-resistant bacteria (**Fig 2F–2I**).

To account for extensive parameter uncertainty reflecting heterogeneity across different bacteria and healthcare facilities, 2 distinct sets of probabilistic Monte Carlo simulations were conducted: (i) generic MRB in generic hospitals; and (ii) case studies of specific bacteria, hospital wards, and COVID-19 response scenarios (see **S1 Appendix section 1.3**). Pathogen transmission rates per unit of time estimated from the literature ($\beta$) were scaled to interindividual contact rates per unit of time ($\kappa$) to approximate transmission rates per unit of contact ($\pi = \beta/\kappa$), facilitating generalizability across settings. Monte Carlo simulations were conducted by randomly sampling parameter values from their respective probability distributions, yielding a distinct parameter vector $\Theta$ for each simulation (see **S1 Appendix section 1.2**). Probability distributions for key parameters underlying these distinct simulation sets are visualized in **Figure C in S1 Appendix**. Bootstrap resampling was used to determine the appropriate number of Monte Carlo simulations to conduct ($n = 500$; **Figures D and E in S1 Appendix**).

### Evaluating impacts of COVID-19 on antibiotic resistance

Epidemiological indicators ($\Gamma$) were calculated from simulation outputs and include the prevalence and incidence of SARS-CoV-2 infection, of patient colonization with $B^S$ and $B^R$, and of HCW carriage of $B^S$ and $B^R$, as well as the cumulative number of patient-days of bacterial colonization and the average resistance rate (the cumulative share of patient colonization due to $B^R$ relative to $B^S$) (see **S1 Appendix section 1.4**). Multivariate sensitivity analyses were conducted to determine which model parameters drive respective $\Gamma$ (**Figures F and G in S1 Appendix**).

Epidemiological impacts of COVID-19 were assessed by calculating how COVID-19 responses impacted epidemiological indicators in parameter-matched simulations. For $\Theta^i$ corresponding to the $i^{th}$ Monte Carlo simulation, the model was run both with selected COVID-19 response parameters ($\tau > 0$) and without ($\tau = 0$), and corresponding epidemiological indicators were calculated in their presence ($\Gamma_1^i$) and absence ($\Gamma_0^i$). The epidemiological change resulting from COVID-19 responses was thus calculated as the relative difference in each indicator,

$$\Delta\Gamma^i = \left(\frac{\Gamma_1^i}{\Gamma_0^i} - 1\right) \times 100\%$$

such that positive (negative) values indicate the percentage increase (decrease) in $\Gamma^i$ as a result of the COVID-19 response parameters included in $\Theta^i$. Final differences for each indicator are reported as means and 95% uncertainty intervals (95% UIs, the 2.5th and 97.5th quantiles) of the resulting distribution $\Delta\Gamma^n$. $P$ values are not reported due to lack of interpretability for simulated data.

## Results

### Impacts of COVID-19 responses on generic MRB in generic hospitals

The first simulation set accounts for broad parameter ranges, representing "generic multidrug-resistant bacteria" (MRB) across "generic hospitals" in the context of COVID-19

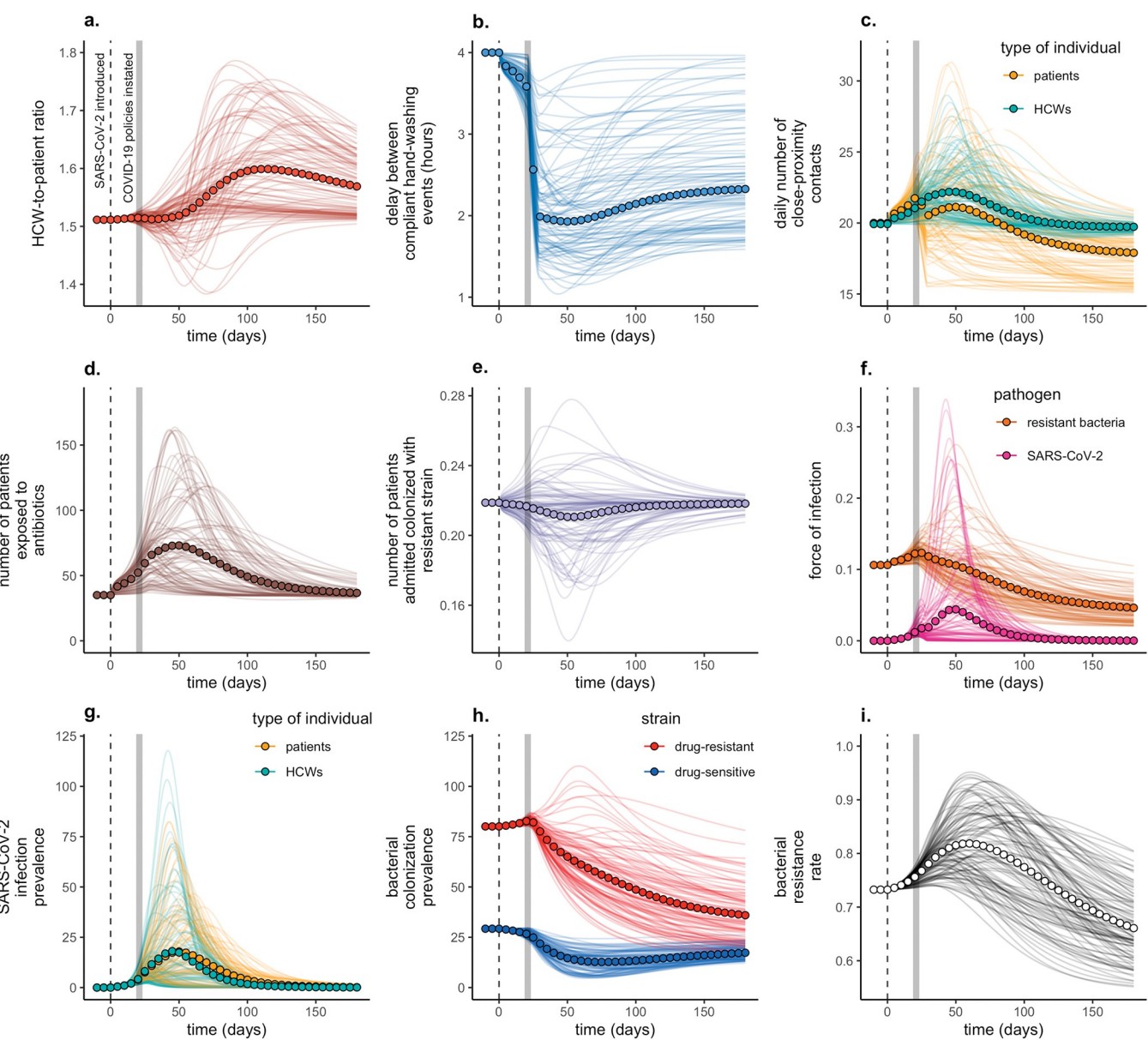

**Fig 2. Combined influence of all COVID-19 responses on epidemiological dynamics of SARS-CoV-2 and a generic commensal bacterium among patients and HCWs in a simulated long-term care hospital in France.** Baseline conditions include 350 beds, a 1.51:1 HCW:patient ratio, 10% antibiotic exposure prevalence ($A_{base} = 0.1$), 40% HCW compliance with hand hygiene after HCW–patient contact ($H_{base} = 0.4$), and an average 80-day patient length of stay ($\mu = 0.0125$). Two index SARS-CoV-2 infections are introduced at $t = 0$ (vertical dashed lines), and policy responses are implemened at $t_{policy} = 21$ days (vertical grey bars) with an intervention burn-in period of $t_{impl} = 7$ days. Lines represent dynamics across $n = 100$ independent simulations, in which all model parameters are fixed except for COVID-19 response parameters ($\tau$), which are drawn randomly ($\tau \sim \mathcal{U}[0, 1]$) for each $\tau$ in each simulation (see **Tables A and B in S1 Appendix**). Circles represent means across all simulations. (**a**) The ratio of HCWs to patients in the hospital ($N^{hcw}/N^{pat}$). (**b**) The average delay between compliant HCW handwashing events ($\omega/day^{-1} \times 24$ hours/day), i.e., the average duration of transient bacterial carriage. (**c**) The average number of contacts that patients have with patients and HCWs ($\kappa^{pat \to pat} + \kappa^{pat \to hcw}$, gold) and that HCWs have with patients and HCWs ($\kappa^{hcw \to hcw} + \kappa^{hcw \to pat}$, green). (**d**) The average number of patients exposed to antibiotics ($A_{SER} \times \sum_g (S_g^{pat} + E_g^{pat} + R_g^{pat}) + A_I \times \sum_g I_g^{pat}$). (**e**) The average number of patients admitted already colonized with resistant bacteria ($\mu^{adm} \times N^{beds} \times f_{C^R}$). (**f**) Forces of infection for SARS-CoV-2 ($\sum_i \lambda_V^i$, magenta) and antibiotic-resistant bacteria ($\sum_i \lambda_{BR}^i$, orange). (**g**) The number of active SARS-CoV-2 infections among patients ($V^{pat}$, gold) and HCWs ($V^{hcw}$, green). (**h**) The number of patients colonized with antibiotic-sensitive bacteria ($C^S$, blue) and antibiotic-resistant bacteria ($C^R$, red). (**i**) The resistance rate, the proportion of colonized patients bearing antibiotic-resistant bacteria [$C^R/(C^S+C^R)$]. COVID-19, Coronavirus Disease 2019; HCW, healthcare worker; SARS-CoV-2, Severe Acute Respiratory Syndrome Coronavirus 2.

responses of intermediate magnitude ($\tau = 0.5$) (see parameter distributions in **Table B in S1 Appendix**). In the absence of COVID-19, these hospitals and MRB are characterized by substantial epidemiological heterogeneity (**Figures H and I in S1 Appendix**).

## Combined COVID-19 responses prevent MRB colonization but favour resistance

When all COVID-19 responses are combined ($\tau = 0.5$), nosocomial SARS-CoV-2 outbreaks have varied impacts on healthcare-associated behaviours, hospital demography, and, consequently, the epidemiological burden of MRB (**Fig 3**). Collateral impacts favouring MRB colonization include increased rates of HCW contact and increased patient exposure to antibiotics.

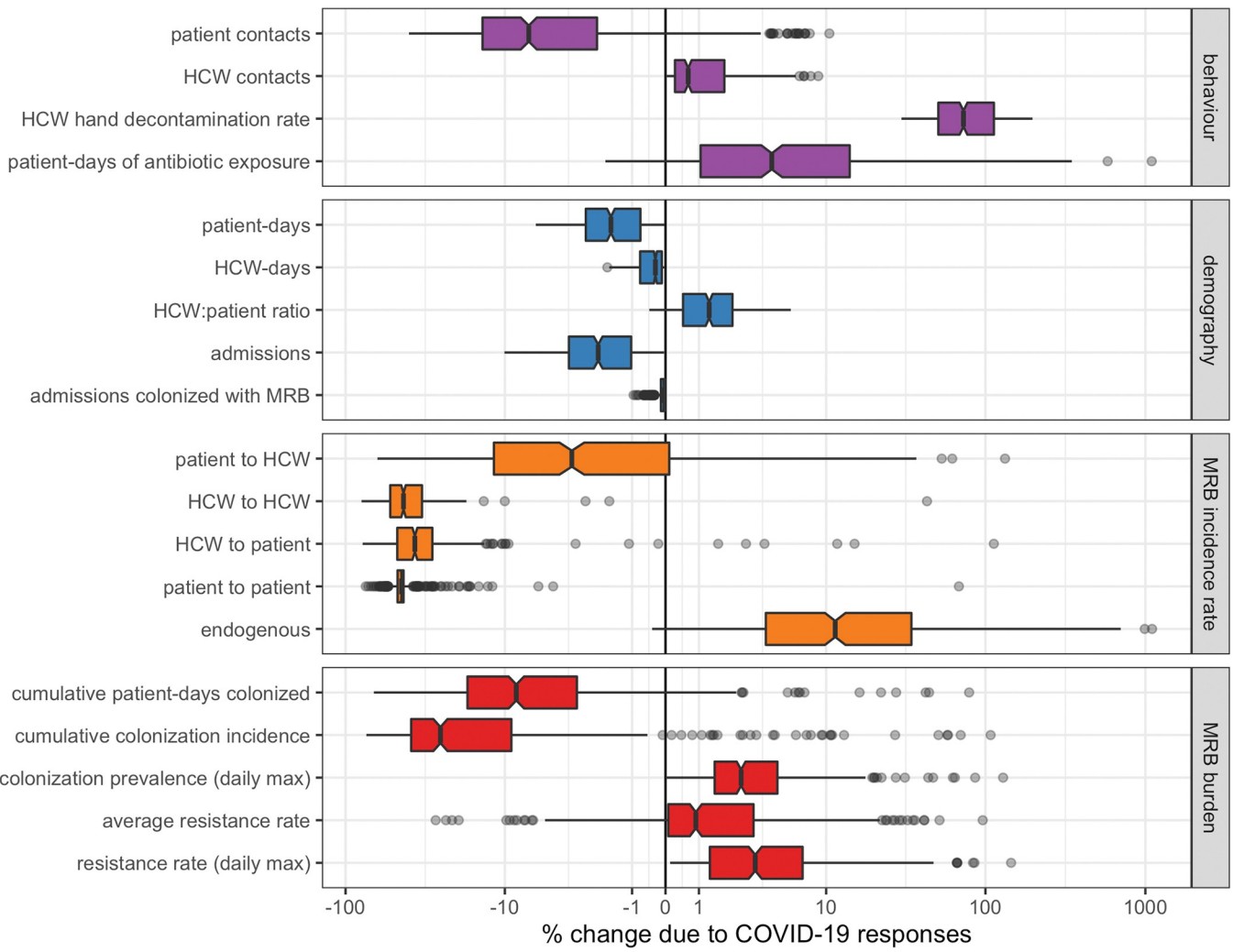

**Fig 3. Combined COVID-19 responses impact the behaviour and demography of generic hospital populations and consequently impact the epidemiological dynamics of generic MRB.** Each data point represents one of $n = 500$ unique MRB–hospital pairs. For each indicator (row), change due to COVID-19 responses is calculated as the difference between parameter-matched simulations including all COVID-19 responses simultaneously ($\tau = 0.5$) versus those including no COVID-19 responses ($\tau = 0$). Indicators are calculated cumulatively over $t = 180$ days of simulation, after introduction of 2 index cases of SARS-CoV-2 into the hospital at $t = 0$. Boxplots represent the $IQR$; whiskers extend to the furthest value up to $\pm 1.5 \times IQR$; and notches extend to $1.58 \times IQR/\sqrt{n}$. Scales are pseudo-log$_{10}$-transformed using an inverse hyperbolic sine function (R package ggallin) [27]. COVID-19, Coronavirus Disease 2019; HCW, healthcare worker; MRB, multidrug-resistant bacteria; SARS-CoV-2, Severe Acute Respiratory Syndrome Coronavirus 2.

Collateral impacts preventing MRB colonization include reduced rates of patient contact, increased rates of HCW hand decontamination, and an increased HCW:patient ratio.

Combined COVID-19 responses result in a mean 88.1% [95% UI: 58.8%, 99.7%] reduction in cumulative nosocomial SARS-CoV-2 infection incidence, including reductions across all acquisition routes (**Figure J in S1 Appendix**), but have more heterogeneous impacts on MRB epidemiology (**Fig 3**). COVID-19 responses lead to a mean 11.6% [−2.7%, 44.1%] reduction in the cumulative number of patient-days colonized with MRB and a mean 24.2% [−7.8%, 59.3%] reduction in the cumulative incidence of nosocomial colonization. However, incidence rates decrease for some acquisition routes (e.g., HCW-to-patient transmission) but increase for others (e.g., endogenous acquisition). COVID-19 responses also lead to transient increases in patient MRB colonization, with a mean 4.7% [0.4%, 18.7%] increase in peak colonization prevalence, as well as a 2.9% [−5.4%, 23.2%] increase in the average resistance rate (the cumulative share of colonized patient-days caused by resistant bacteria).

## Distinct COVID-19 responses have distinct epidemiological impacts

Individual COVID-19 responses have distinct impacts on MRB epidemiology (**Fig 4A**). Several COVID-19 responses are responsible for reducing the cumulative number of patient-days of MRB colonization, including reduced patient admission ($\tau_{ra}$), improved HCW hand hygiene ($\tau_{hh}$), and patient lockdown ($\tau_{pl}$). However, no COVID-19 responses lead to meaningful reductions in the average resistance rate, although hand hygiene and patient lockdown are associated with high variance in this outcome despite negligible change on average. The COVID-19 response that most favours increasing resistance is abandoned stewardship ($\tau_{as}$), which, in the absence of other COVID-19 responses, causes a mean 9.4% [0.1%, 49.0%] increase in the average resistance rate. This is followed by sicker casemix ($\tau_{sc}$) with a 2.5% [<0.1%, 7.7%] increase in the average resistance rate, and COVID-19 prescribing ($\tau_{cp}$) with a 1.0% [<0.1%, 6.5%] increase.

Individual COVID-19 responses also have distinct impacts on the incidence of SARS-CoV-2 infection and MRB colonization, with changes varying across different routes of acquisition. Reductions in SARS-CoV-2 incidence are led by reduced transmission from all individuals due to universal masking ($\tau_{um}$), reduced transmission from HCWs due to staff sick leave ($\tau_{ss}$), reduced transmission to patients due to reduced admissions ($\tau_{ra}$), and reduced patient-to-patient transmission due to patient lockdown ($\tau_{pl}$) (**Figure K in S1 Appendix**). Reductions in MRB incidence are largely due to reduced transmission from HCWs as a result of improved hand hygiene ($\tau_{hh}$), reduced transmission from patients due to patient lockdown ($\tau_{pl}$), and reduced transmission to and from all individuals due to reduced patient admission ($\tau_{ra}$) (**Figure L in S1 Appendix**). These COVID-19 responses outweigh the impacts of competing COVID-19 responses that favour greater SARS-CoV-2 incidence [care disorganization ($\tau_{cd}$) and COVID-19 stays ($\tau_{cs}$)] and/or MRB incidence [abandoned stewardship ($\tau_{as}$), care disorganization ($\tau_{cd}$), sicker casemix ($\tau_{sc}$), COVID-19 stays ($\tau_{cs}$), staff sick leave ($\tau_{ss}$), and COVID-19 prescribing ($\tau_{cp}$)] (**Figures M and N in S1 Appendix**).

## Surges in COVID-19 caseloads lead to strong selection for resistance

Impacts of COVID-19 responses on MRB epidemiology vary across policy responses and caseload responses (**Fig 4B**). The cumulative number of patient-days colonized with MRB decreases by a mean 13.2% [0.6%, 46.9%] with policy responses, but increases by a mean 6.3% [−5.2%, 42.8%] with caseload responses. Mirroring these trends, MRB colonization incidence decreases by 28.2% [2.5%, 60.2%] with policy responses but increases by 13.8% [−3.5%, 77.0%] with caseload responses (**Figure M in S1 Appendix**). By contrast, policy responses have little

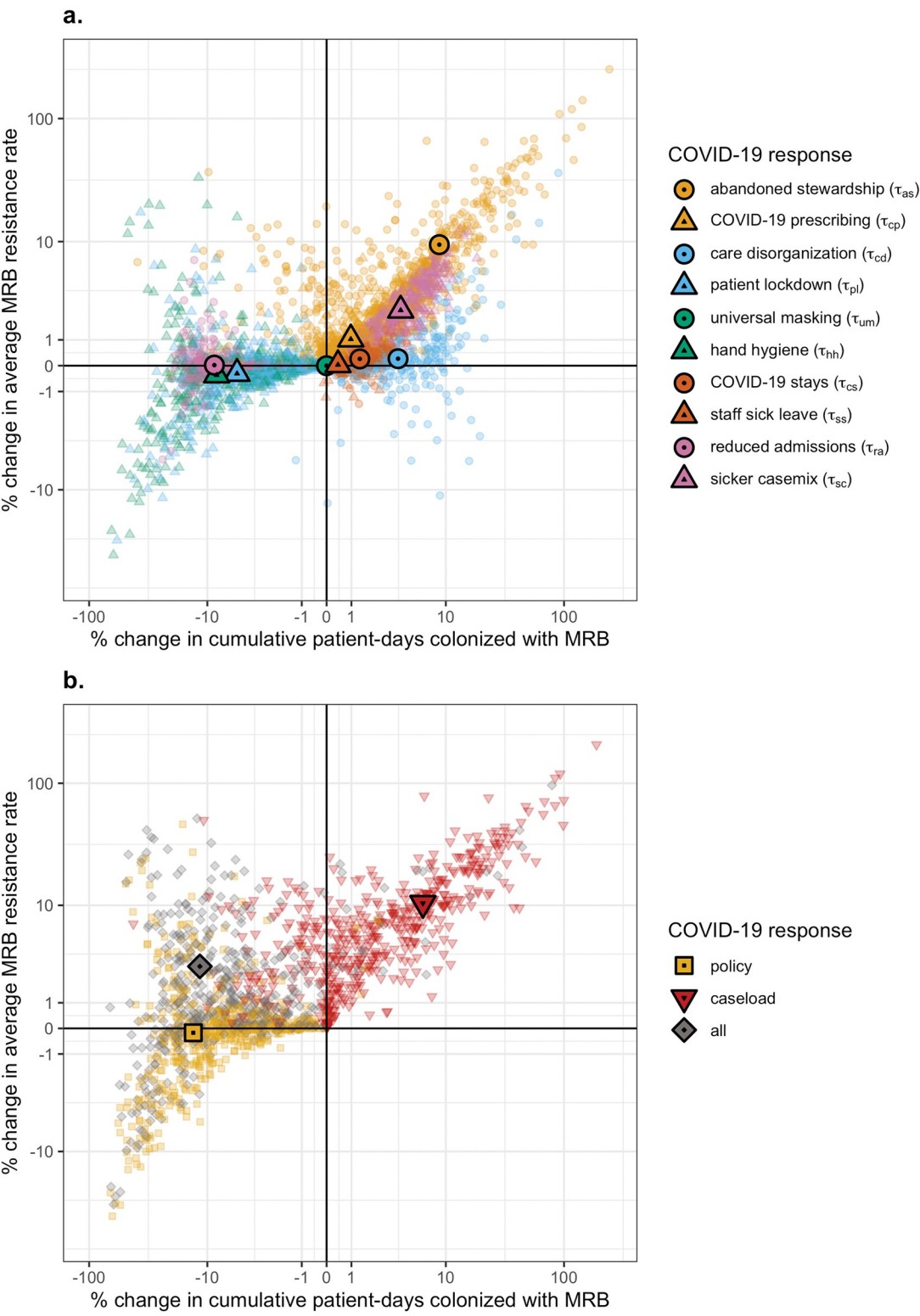

**Fig 4. Some COVID-19 responses prevent, while others favour MRB colonization.** Scatter plots depict change in the cumulative number of patient-days of MRB colonization (x-axis) and the average MRB resistance rate (y-axis) resulting from (**a**) individual COVID-19 responses, each given by a unique colour–shape combination, and (**b**) combinations of COVID-19 responses. Policy responses include COVID-19 prescribing ($\tau_{cp}$), patient lockdown ($\tau_{pl}$), universal masking ($\tau_{um}$), and hand hygiene ($\tau_{hh}$). Caseload responses include abandoned stewardship ($\tau_{as}$), care disorganization ($\tau_{cd}$), COVID-19 stays ($\tau_{cs}$), staff

sick leave ($\tau_{ss}$), reduced admission ($\tau_{ra}$), and sicker casemix ($\tau_{sc}$). Small translucent points represents unique MRB–hospital pairs, and larger opaque points represent means across $n$ =500 pairs. For each indicator, change due to COVID-19 responses is calculated as the difference between parameter-matched simulations including respective COVID-19 responses ($\tau$ =0.5) versus those including no COVID-19 responses ($\tau$ = 0). Indicators are calculated cumulatively over $t$ = 180 days of simulation, after introduction of 2 index cases of SARS-CoV-2 into the hospital at $t$ =0. Scales are pseudo-$\log_{10}$-transformed using an inverse hyperbolic sine function (R package ggallin). COVID-19, Coronavirus Disease 2019; MRB, multidrug-resistant bacteria; SARS-CoV-2, Severe Acute Respiratory Syndrome Coronavirus 2.

impact on the average resistance rate, while caseload responses lead to a mean 10.4% [0.2%, 46.9%] increase.

## Impacts of COVID-19 on MRSA and ESBL-*E. coli* across wards and scenarios

The second simulation set accounts for a series of case studies representing specific bacteria (MRSA and ESBL-*E. coli*; see **Table C in S1 Appendix**), specific healthcare settings (a geriatric rehabilitation ward, a short-stay geriatric ward, and a general paediatric ward; see **Table D in S1 Appendix**), and specific COVID-19 response scenarios (an organized response, an intermediate response, and an overwhelmed response; see **Table E in S1 Appendix**). Response scenarios were assumed to vary in terms of the COVID-19 policies put in place. For example, in the organized response, we assumed universal masking with N95 respirators, strict patient lockdown, a large increase in hand hygiene compliance, and COVID-19 prescribing only for patients experiencing bacterial coinfection; and in the disorganized response, we assumed no masking, minimally effective patient lockdown, marginal improvement in hand hygiene, and high rates of COVID-19 prescribing.

## Baseline nosocomial dynamics differ across wards

Baseline healthcare-associated behaviours and demography vary across wards (**Figure O in S1 Appendix**), resulting in ward-specific epidemiological dynamics of MRSA and ESBL-*E. coli* colonization (prior to introduction of SARS-CoV-2) (**Figure P in S1 Appendix**). Due to different factors including ward-specific differences in antibiotic exposure, patient length of stay, and interindividual contact rates, the relative importance of different colonization acquisition routes varies across settings and bacteria. For MRSA, for example, endogenous acquisition dominates in the short-stay ward, HCW-to-patient transmission dominates in the general ward, and patient-to-patient transmission dominates in the rehabilitation ward (**Figure Q in S1 Appendix**). Upon introduction of SARS-CoV-2, ward-specific characteristics further translate to variability in SARS-CoV-2 risk and infection dynamics (**Figure R in S1 Appendix**). In both the short-stay ward and the general ward, most SARS-CoV-2 transmission results from HCWs, with negligible transmission from patients to HCWs or other patients. Conversely, in the rehabilitation ward, patient-to-patient transmission is the dominant acquisition route.

## Overwhelmed COVID-19 responses exacerbate antibiotic resistance

Impacts of SARS-CoV-2 outbreaks on antibiotic resistance epidemiology vary across wards, bacterial species, and COVID-19 response scenarios (**Fig 5**). SARS-CoV-2 outbreaks have little impact on bacteria acquired predominantly via endogenous acquisition, including ESBL-*E. coli* in the general ward and both MRSA and ESBL-*E. coli* in the short-stay ward (**Figure Q in S1 Appendix**). For remaining contexts with substantial bacterial transmission to and from patients, SARS-CoV-2 outbreaks have significant impacts on bacterial colonization incidence and the bacterial resistance rate (**Fig 5**). Overwhelmed COVID-19 responses are associated

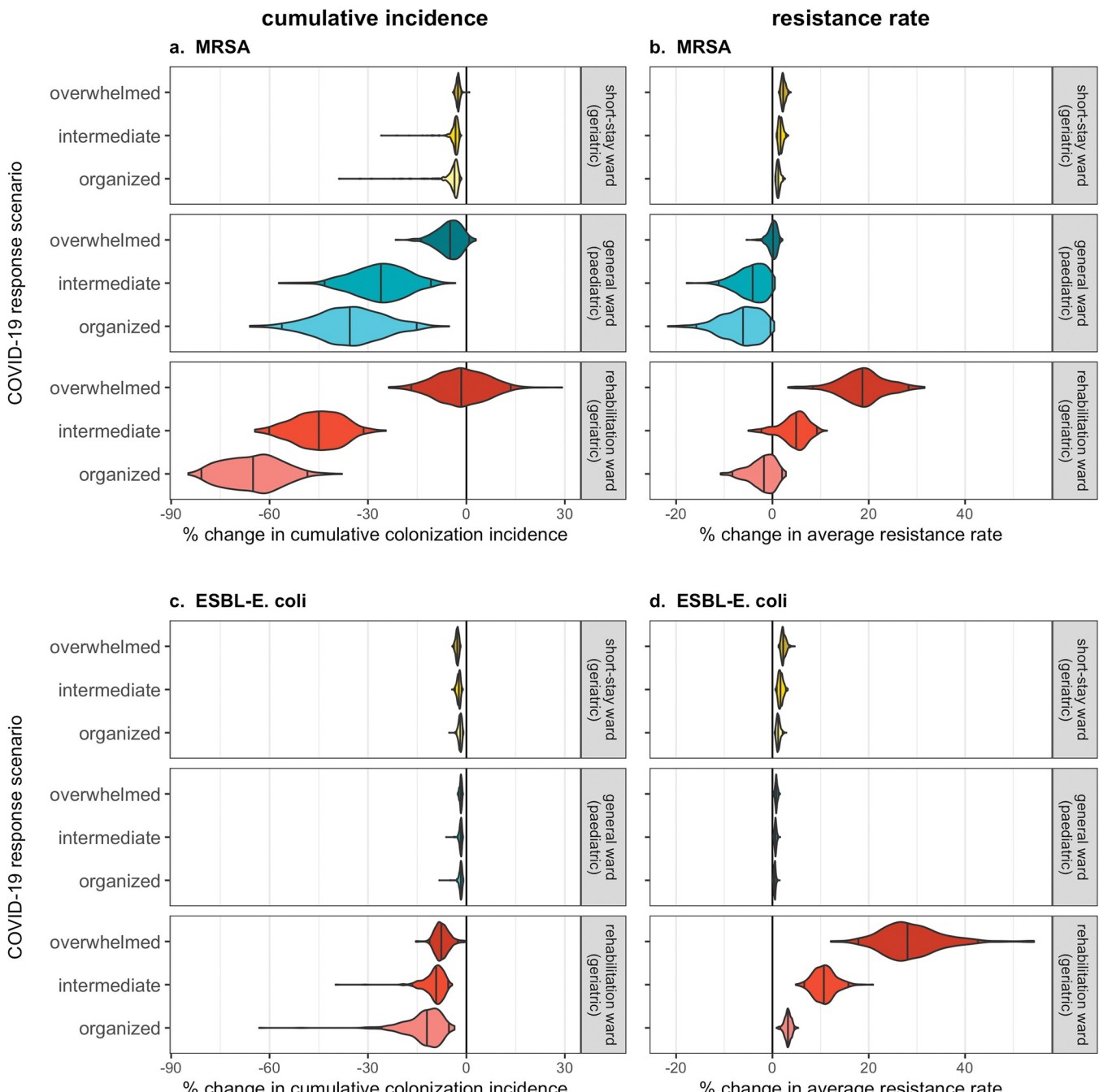

**Fig 5. Overwhelmed responses to COVID-19 result in greater colonization burden of antibiotic-resistant bacteria relative to organized responses.** Violin plots represent outcome distributions from $n = 500$ Monte Carlo simulations and depict cumulative change in epidemiological indicators due to nosocomial SARS-CoV-2 outbreaks (x-axis) across different COVID-19 response scenarios (y-axis). Results are presented for (**a**) cumulative MRSA colonization incidence, (**b**) the average MRSA resistance rate, (**c**) cumulative ESBL-*E. coli* colonization incidence, and (**d**) the average ESBL-*E. coli* resistance rate. For each hospital ward, bacterial species, and COVID-19 response scenario, change due to COVID-19 responses is calculated as the difference between parameter-matched simulations including respective COVID-19 responses (organized, intermediate, or overwhelmed; see **Table E in S1 Appendix**) versus those including no COVID-19 responses ($\tau = 0$), assuming baseline values of SARS-CoV-2 transmissibility ($\beta_V = 1.28$) and policy implementation timing ($t_{policy} = 21$). Indicators are calculated cumulatively over $t = 180$ days of simulation, after introduction of 2 index cases of SARS-CoV-2 into the hospital at $t = 0$. COVID-19, Coronavirus Disease 2019; ESBL, extended-spectrum beta-lactamase-producing; MRSA, methicillin-resistant *Staphylococcus aureus*; SARS-CoV-2, Severe Acute Respiratory Syndrome Coronavirus 2.

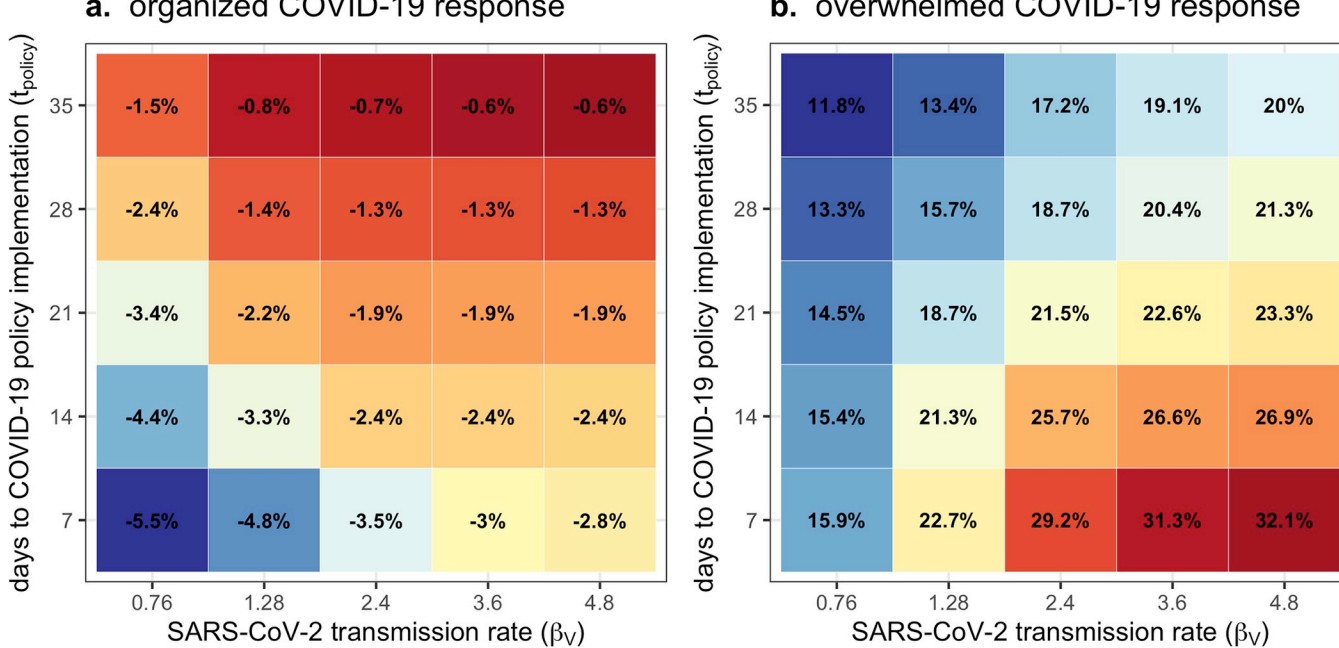

**Fig 6. Antibiotic resistance is mitigated by earlier implementation of organized COVID-19 responses but exacerbated by earlier implementation of overwhelmed COVID-19 responses.** Each coloured tile depicts mean change in the average resistance rate of MRSA across $n = 500$ Monte Carlo simulations, which varies with the SARS-CoV-2 transmission rate (x-axis) and the delay to COVID-19 policy implementation (y-axis) in a geriatric rehabilitation ward with (**a**) an organized response to COVID-19 versus (**b**) an overwhelmed response. Change due to COVID-19 responses is calculated as the difference between parameter-matched simulations including respective COVID-19 responses (organized, overwhelmed; see **Table E in S1 Appendix**) versus those including no COVID-19 responses ($\tau = 0$). Average resistance rate is calculated cumulatively over $t = 180$ days of simulation, after introduction of 2 index cases of SARS-CoV-2 into the hospital at $t = 0$. COVID-19, Coronavirus Disease 2019; MRSA, methicillin-resistant *Staphylococcus aureus*; SARS-CoV-2, Severe Acute Respiratory Syndrome Coronavirus 2.

with higher colonization incidence and higher resistance rates than organized responses. For example, given an organized response in the rehabilitation ward, colonization incidence of MRSA decreases by a mean 65.2%% [48.8%, 79.9%], with little change in the resistance rate. Conversely, given an overwhelmed response, there is little change in incidence, while the resistance rate increases by a mean 18.7% [8.5%, 28.3%]. These impacts result from how different COVID-19 response scenarios modify healthcare-associated behaviours and demography in each ward. In the rehabilitation ward, for example, patient antibiotic exposure and the daily number of contacts per patient tend to increase in the overwhelmed scenario but decrease in the organized scenario (**Figure S in S1 Appendix**).

Impacts of COVID-19 on antibiotic resistance also depend on the transmissibility of SARS-CoV-2 ($\beta_V$) and the timing of COVID-19 policy implementation ($t_{policy}$) (visualized for MRSA in the rehabilitation ward in **Fig 6**; see all bacteria and wards in **Figures T and U in S1 Appendix**). Increasing the SARS-CoV-2 transmission rate results in larger SARS-CoV-2 outbreaks and, in turn, greater selection for resistance across all wards, bacteria, and control scenarios. However, impacts of policy timing depend on the nature of the policies being implemented. Earlier implementation of organized responses generally results in lower resistance rates, due to their ability to help control the spread of and selection for resistant bacteria. Conversely, earlier implementation of overwhelmed responses generally results in higher resistance rates, as these responses tend to exert additional selection for resistant bacteria.

## Discussion

This study demonstrates how collateral impacts of COVID-19 may both favour and prevent against the spread of antibiotic resistance in healthcare settings. Surges in COVID-19 cases—and associated consequences like abandonment of antibiotic stewardship programmes and disorganization of patient care—were found to favour the spread of resistant bacteria. Conversely, COVID-19 control policies like patient lockdown, universal masking, and reinforcement of hand hygiene were effective for prevention of bacterial colonization. Such policies work not only by directly preventing bacterial transmission, but also by limiting surges in COVID-19 cases and the conditions favourable for bacterial spread that they create. These findings thus suggest that limiting the proliferation of antibiotic resistance is an important collateral benefit of nosocomial COVID-19 prevention. This further suggests that various other public health strategies effective for prevention of SARS-CoV-2 transmission in healthcare settings—including vaccination, mass testing, and HCW cohorting—may help to alleviate the spread of antibiotic resistance [28–30].

Findings also suggest that better pandemic preparedness may serve to limit unintended selection for antibiotic-resistant bacteria. Our simulations tailored to an early pandemic context found that patients in better organized healthcare facilities (i.e., those enacting more effective COVID-19 control policies sooner) were less likely to acquire bacterial colonization and experienced lower rates of resistance than patients in facilities overwhelmed by COVID-19. We further found that facilities with higher underlying rates of bacterial transmission were at greatest risk of pandemic-associated surges in antibiotic resistance. In the context of poorly anticipated outbreaks of any novel respiratory pathogen, healthcare facilities that rapidly instate effective IPC measures and maintain stable staffing ratios while limiting unnecessary surges in antibiotic use may thus avoid concomitant surges in antibiotic resistance. However, putting such an organized response into action depends importantly on the financial and human resources available and on the emergency management systems already in place. With hindsight, the rapid global spread of SARS-CoV-2 and its associated health system shocks in early 2020 revealed insufficient global capacity to detect and contain novel pathogens with pandemic potential [31]. This has spurred calls for transformational change in international law and governance, and expansive global investment in pandemic preparedness [32]. Our study suggests that mitigating the spread of antibiotic-resistant bacteria should be considered as a collateral benefit of pandemic preparedness initiatives, with implications for their funding and design.

Collateral impacts of COVID-19 have evolved over successive pandemic waves and will continue to evolve through the transition to endemic COVID-19 [33,34]. Such variability has coincided with the ebbing and flowing of enforcement of nonpharmaceutical COVID-19 control interventions, availability of vaccines and antiviral therapies, and capacity and resilience of healthcare systems. The clinical and epidemiological characteristics of COVID-19 are also in constant flux, due to evolution of intrinsic virulence and immune escape properties of SARS-CoV-2 variants, and great heterogeneity in acquisition and waning of both natural and vaccine-induced immunity. Understanding how these and other COVID-19–related factors combine to influence the spread of antibiotic-resistant bacteria remains a great challenge. Asymptomatic bacterial carriage is difficult to detect and relatively rarely monitored, and interrupted epidemiological surveillance and the reallocation of public health resources away from antibiotic resistance programmes and towards COVID-19 control have made MRB surveillance particularly challenging. In many instances, SARS-CoV-2 testing and surveillance infrastructure has been repurposed from existing antimicrobial resistance infrastructure, leading to extensive gaps and delays in the reporting of antimicrobial resistance data since the

onset of the pandemic, and a reduction in the number of bacterial isolates sent for whole genome sequencing [7]. Further, data underlying the causal pathways proposed to link the epidemiological dynamics of COVID-19 and MRB (e.g., impacts of the pandemic on contact behaviour, IPC compliance, care delivery pathways, and underlying rates of bacterial colonization and resistance) are sorely lacking.

In light of such complexity and data limitations, mathematical modelling is a powerful tool to help better understand and disentangle the complex, overlapping mechanisms linking COVID-19 and antibiotic resistance epidemiology. In this context, our model proposes theoretical explanations as to how outbreaks of an emerging pandemic pathogen like SARS-CoV-2 may be expected to exacerbate the spread of antibiotic resistance, and how the timely implementation of effective control measures can mitigate these impacts. Where data are lacking, mathematical modelling should continue to be exploited as a tool to understand mechanistic links between COVID-19 and antibiotic resistance beyond the early pandemic context explored here. Future work should evaluate impacts across other types of healthcare and residential facilities (e.g., retirement homes and prisons), nosocomial bacteria (e.g., *Pseudomonas aeruginosa*, *Acinetobacter baumannii*, and *Enterobacter* spp.), and specific medical procedures associated with both COVID-19 and nosocomial MRB spread (e.g., mechanical ventilation and central venous catheterization). It would also be helpful to evaluate impacts of variable availability of key resources, including masks, diagnostic tests, personal protective equipment, and appropriately trained medical staff.

Future work is also needed to understand impacts of COVID-19 on antibiotic resistance across entire health systems and in community settings. An international surge in outpatient antibiotic consumption was observed initially in March 2020, associated primarily with antimicrobials frequently prescribed to COVID-19 patients early in the pandemic (e.g., azithromycin and hydroxychloroquine) [35,36]. Subsequently, there was an estimated 19% reduction in global antimicrobial consumption from April to August 2020 (relative to 2019) [37]. In combination with reduced human mobility, contact rates, and care-seeking during COVID-19 lockdowns, epidemiological impacts of modified antibiotic consumption in the community remain poorly understood [15]. Data subsequent to first wave lockdowns have shown a decrease in ESBL resistance among *E. coli* isolates in France [38], and reduced carriage of cephalosporin- and carbapenem-resistant Enterobacterales in Botswana [39]. However, more longitudinal estimates from diverse geographical regions and bacterial species are greatly needed.

Although few studies have reported explicitly on impacts of COVID-19 on distributions of antibiotic-resistant strains or serotypes, COVID-19 lockdowns in early 2020 clearly reduced incidence of disease due to community-associated respiratory bacteria like *S. pneumoniae*, *Haemophilus influenzae*, and *Neisseria meningitidis* [3,40–43]. Yet over the same time period, intriguingly, emerging data report persistent carriage of *S. pneumoniae* in the community across countries and age groups [44–46]. It has been suggested that reduced incidence of bacterial infection may thus be explained at least in part by concomitant prevention of other respiratory viruses like influenza [47], which have been shown to favour progression from bacterial colonization to disease [48]. Fully understanding impacts of COVID-19 on any particular form of antibiotic resistance may therefore require taking into account not only SARS-CoV-2 and the bacterium in question, but also other interacting microorganisms. For simplicity, and due to limited evidence of a strong association between SARS-CoV-2 and bacterial coinfection [18], our model assumes no impact of SARS-CoV-2 infection on bacterial acquisition, growth, transmission, or clearance. Yet the extent to which SARS-CoV-2 is prone to within-host virus–virus and/or virus–bacteria interactions remains relatively unclear, may continue to evolve, and could have important consequences for epidemiological dynamics and clinical manifestations of antibiotic resistance [49].

Our model should be considered in the context of several limitations and simplifying assumptions. First, although impacts of COVID-19 on hospital admissions are accounted for, we do not explicitly model community dynamics, nor do we explore different scenarios of SARS-CoV-2 importation from the community. Yet community SARS-CoV-2 outbreaks also result in surges in COVID-19 hospitalizations and the potential overwhelming of healthcare services and have likely played an important role in driving selection for antibiotic-resistant bacteria since the onset of the pandemic. In the context of prohibited hospital visitation during the first wave of COVID-19, staff but not patient interactions with the community—and potential acquisition of both SARS-CoV-2 infection and bacterial carriage—may further be important drivers of nosocomial transmission dynamics. Second, HCWs are conceptualized here as transient vectors, but HCW colonization can also impact transmission dynamics. In particular, nares are a key site for MRSA colonization [50], and chronic HCW colonization has been found to drive prolonged nosocomial MRSA outbreaks [51]. Further, our model is conceptualized as applying to commensal bacteria spread through contact and fomites, so we conservatively assumed that face masks have no impact on bacterial transmission. However, respiratory droplets may play a nonnegligible role, particularly for MRSA transmission [52], so impacts of COVID-19 responses on MRSA colonization incidence may be underestimated. Bacterial strains are also conceptualized as competing exclusively for hosts, although co-colonization is widely observed *in vivo*. However, exclusive colonization is a common modelling approach for practical reasons (to keep models as simple as possible) and due to limited data describing within-host ecological competition dynamics [53]. Finally, our deterministic modelling approach does not allow for stochastic effects like epidemiological extinctions, which are particularly relevant in small populations like hospital wards.

In conclusion, this work has helped to disentangle how nosocomial SARS-CoV-2 outbreaks influence the epidemiological dynamics of MRB. Our simulation-based approach facilitated the exploration of diverse scenarios and broad parameter spaces, helping to unravel the complexity and context specificity of such impacts. Results suggest that surges in antibiotic resistance may be expected as a collateral impact of sudden nosocomial outbreaks of novel respiratory pathogens but that effective implementation of IPC policies that limit nosocomial transmission can mitigate selection for resistance. Given the persistence of SARS-CoV-2 transmission in human populations and high risk of future zoonotic spillovers of other pathogens with pandemic potential [54], investment in outbreak preparedness should be considered a crucial element in the fight against antibiotic resistance.

## Supporting information

**S1 Appendix.** File containing detail on model structure (**section 1.1; Figures A and B**); a description of model simulations (**section 1.2**); a description of model parameterization (**section 1.3; Tables A–E; Figure C**); detail on calculation of methodological indicators (**section 1.4**); simulation convergence and sensitivity analyses (**section 1.5; Figures D–G**); supplementary results for generic MRB in generic hospitals (**section 1.6; Figures H–N**); and supplementary results for case studies of specific bacteria in specific hospital wards (**section 1.7; Figures O–U**).
(PDF)

## Acknowledgments

We thank the members of the EMAE-MESuRS Working Group on Nosocomial SARS-CoV-2 Modelling for helpful discussion. We are also grateful for material support provided by the

French National Institute for Health and Medical Research (Inserm), Institut Pasteur, le Conservatoire National des Arts et Métiers, and l'Université Versailles Saint-Quentin-en-Yvelines/ Université Paris-Saclay.

## Author Contributions

**Conceptualization:** David R. M. Smith, Laura Temime, Lulla Opatowski.

**Formal analysis:** David R. M. Smith.

**Funding acquisition:** Laura Temime, Lulla Opatowski.

**Investigation:** David R. M. Smith, George Shirreff.

**Software:** David R. M. Smith.

**Supervision:** Laura Temime, Lulla Opatowski.

**Visualization:** David R. M. Smith.

**Writing – original draft:** David R. M. Smith.

**Writing – review & editing:** David R. M. Smith, George Shirreff, Laura Temime, Lulla Opatowski.

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
