## [Editor Report · Decision Letter 0]

27 Sep 2022

Dear Dr Smith, 

Thank you for submitting your manuscript entitled "Collateral impacts of pandemic COVID-19 drive the nosocomial spread of antibiotic resistance: a modelling study" for consideration by PLOS Medicine.

Your manuscript has now been evaluated by the PLOS Medicine editorial staff as well as by an academic editor with relevant expertise and I am writing to let you know that we would like to send your submission out for external peer review.

Please re-submit your manuscript within two working days, i.e. by Sep 29 2022 11:59PM.

Kind regards,

Philippa Dodd, MBBS MRCP PhD

PLOS Medicine

---

## [Decision Letter · Decision Letter 1]

14 Feb 2023

Dear Dr. Smith,

Thank you very much for submitting your manuscript "Collateral impacts of pandemic COVID-19 drive the nosocomial spread of antibiotic resistance: a modelling study" (PMEDICINE-D-22-03098R1) for consideration at PLOS Medicine. 

Your paper was evaluated by a senior editor and discussed among all the editors here. It was also sent to independent reviewers, including a statistical reviewer. The reviews are appended at the bottom of this email and any accompanying reviewer attachments can be seen via the link below:

[LINK]

In light of these reviews, I am afraid that we will not be able to accept the manuscript for publication in the journal in its current form, but we would like to consider a revised version that addresses the reviewers' and editors' comments. Obviously we cannot make any decision about publication until we have seen the revised manuscript and your response, and we plan to seek re-review by one or more of the reviewers. 

We expect to receive your revised manuscript by Mar 07 2023 11:59PM. Please email us (plosmedicine@plos.org) if you have any questions or concerns.

We look forward to receiving your revised manuscript. 

Sincerely,

Philippa Dodd, MBBS MRCP PhD

PLOS Medicine

plosmedicine.org

GENERAL

Please respond to all editor and reviewer comments detailed below, in full

Please remove the statements detailed at lines 26, 28 and 30 and include this information only in the manuscript submission form.

ABSTRACT

Abstract Background: Please ensure that the final sentence clearly states the study question.

Abstract Methods and Findings: as written precise details are rather vague and the text reads more like an introduction rather than a clear description of what you did and what you found. We appreciate that is difficult to describe full details of the models/simulations concisely but we suggest some revision of this section in mind of the below points.

Please use past tense i.e. were not are, for example - line 43 suggest “We conducted a mathematical modelling study to investigate…” or something similar, and line 48 suggest “…model simulations were conducted…”

It may be helpful to include come more specific details, if possible, of (at least some) model parameters, the different hospital settings simulated, varying ratios of patients and healthcare staff, proportions with COVID etc such that the reader gains some contextual insights

How did you decide which “diverse bacterial species” to investigate/simulate, can you give some examples of the major players and justifications perhaps

Please clearly define the main outcome measures for the reader

PLOS Medicine requires that the main results are quantified with 95% CIs and p values. We understand the use of 95% UIs in a study of this design – please include these when reporting outcomes. Please see below also.

STATISTICAL REPORTING

We note the reporting of negative values but also the use of hyphens to separate upper and lower bounds. Suggest the use of commas instead to avoid any confusion between hyphens and negative values. 

As above, PLOS Medicine requires that the main results are quantified with 95% CIs and p values, including in the abstract. We understand and accept the use of 95% UIs in a study of this design. When reporting p values please report as p<0.001 or where higher, as p=0.002, for example. If not reporting p values, for the purpose of transparent data reporting please clearly state the reasons why not. 

AUTHOR SUMMARY

At this stage, we ask that you include a short, non-technical Author Summary of your research to make findings accessible to a wide audience that includes both scientists and non-scientists. The Author Summary should immediately follow the Abstract in your revised manuscript. This text is subject to editorial change and should be distinct from the scientific abstract. 

It may help you to review some examples from published papers on our website here: 

https://journals.plos.org/plosmedicine/

Please see our author guidelines for more information: https://journals.plos.org/plosmedicine/s/revising-your-manuscript#loc-author-summary

INTRODUCTION

Line 100: “Findings demonstrate….” Please either remove or move this sentence to the discussion section of the manuscript, as it reports the study outcomes.

METHODS and RESULTS

We ask that the following details [derived from Geoffrey P Garnett, Simon Cousens, Timothy B Hallett, Richard Steketee, Neff Walker. Mathematical models in the evaluation of health programmes. (2011) Lancet DOI:10.1016/S0140-6736(10)61505-X.] be included with all modelling studies. We think that that you have included all relevant information but please review the list below and amend where necessary:

1) Please provide a diagram that shows the model structure, including how the disease natural history is represented, the process and determinants of disease acquisition, and how the putative intervention could affect the system.

2) Please provide a complete list of model parameters, including clear and precise descriptions of [the meaning of each parameter, together with the values or ranges for each, with justification or the primary source cited, and important caveats about the use of these values noted].

3) Please provide a clear statement about how the model was fitted to the data [including goodness-of-fit measure, the numerical algorithm used, which parameter varied, constraints imposed on parameter values, and starting conditions].

4) For uncertainty analyses, please state the sources of uncertainties quantified and not quantified [can include parameter, data, and model structure].

5) Please provide sensitivity analyses to identify which parameter values are most important in the model. Uncertainty estimates seek to derive a range of credible results on the basis of an exploration of the range of reasonable parameter values. The choice of method should be presented and justified.

6) Please discuss the scientific rationale for this choice of model structure and identify points where this choice could influence conclusions drawn. Please also describe the strength of the scientific basis underlying the key model assumptions.

FIGURES

Please consider avoiding the use of green and/or red to ensure that your figures are accessible to those with color blindness

Please ensure that all abbreviations are defined in an appropriate caption for example HCW in figure 2

Figure 3: in the caption you detail confidence interval but in the text you detail uncertainty interval , please clarify/revise accordingly

REFERENCES

Please see our website for reference guidelines: 

https://journals.plos.org/plosmedicine/s/submission-guidelines#loc-references

Please ensure that for in-text reference callouts, citations are placed in square brackets preceding punctuation, for example line 72 should read “…including sexually transmitted infections (e.g. HIV) [1], vector-borne illnesses…” please check and amend throughout where relevant.

Please ensure that the bibliography uses journal name abbreviations found in the National Center for Biotechnology Information (NCBI) databases. 

Please ensure that up to but no more than 6 author names are listed followed by et al, in the event that more than 6 authors contribute to a study.

Please amend where necessary including the supporting files

Comments from the reviewers:

Reviewer #1: See attachment

Michael Dewey

Reviewer #2: The paper reports the results of a modelling study aimed at estimating the impact of COVID-19 pandemic on mutidrug resistant bacteria transmission, in general and according to the type of bacteria, healthcare setting and timelinees and quality of response to COVID-19.

The issue is relevant due to lack of conclusive data on the estimated impact of COVID-19 on antimicrobial resistance epidemiology.

However, the strenght of the study results are highly dependant on the validity of assumptions made for building the study model.

The model is based on parameters accounting for policy response and for caseload responses.

Two observations:

1. one of the parameters is "COVID prophylaxis", accounting for patient exposure to antibiotics for "prevention" of bacterial co-morbidities. 

Antibiotic treatment is not given in the COVID population as prophylaxis, but as empirical therapy when a bacterial co-infection is suspected (because of x-ray or lab parameters suggestive of a co-existing bacterial pneumonia or in all critical patients given the diagnosis' challenge). Thus, the term prophylaxis is wrong and forecasting that all symptomatic patients receive antibiotics is a great overestimation of what should be the appropriate protocol for treament of these patients.

2. Among the parameters used, the authors included the "abandoned stewardship" to take into account the increase in patients exposed to antibiotics due to reduction in antibiotic stewardship activities. A similar parameter has not been included for infection control and prevention. Hand hygiene is surely a core measure to prevent transmission but it is not the only one. Several studies have demonstrated an increase in the incidence of healthcare infections during the pandemic surges (eg the NHSN surveillance report). During the pandemic surge, infection control measures other than hand hygiene and DPI were not consistenlty applied. For example, healthcare workers did not have the time to change all the DPI (including isolation gowns, coveralls, protective suits) between patients and this has increased the likelihood of MRB transmission; similarly, it was frequently reported a lower level of compliance with infection control measures when caring for invasive devices. The incapacity to mantain the infection control level previously reached in several hospitals have been reported as the driver of the observed increase in healthcare associated infections and, given that 2/3 of healthcare infections are due to MRB, the two phenomena are closely linked.

The lack of accounting for a decreased capacity of infection control and prevention is an important study vulnus. 

It would be necessary to change the parameter "COVID prophylaxis" and to include a proxy accounting for the abandoned infection control program.

The study shows that COVID-19 response lead to a reduction of MRB trasmission, while we know that, where data are available (NHSN), an increase of HAIs has been observed (see Baker et al. The Impact of COVID-19 on Healthcare-Associated Infections. Clin Infect Dis accepted paper).

Reviewer #3: The submitted study is based on modelling showing the impact of COVID-19 and related containment actions on the spread of bacterial resistance in the hospitals. Although the evaluation has some limitations recognized by the authors, it provides interesting and useful insights on the effect of the control measures, taking into consideration the characteristics of the contexts and the way the measures are implemeted.

Considering the methodology used and the interest of the results in a public health perspective, in my opinion, the paper can be published with minor changes. The following suggestions are for the attention of the authors who can decide whether to take them into account or not:

- The ecological competition between resistant and antibiotic-susceptible bacteria is not always proven and in any case appears not to be complete. For example, according to some authors MRSA does not replace but goes (at least in part) in addition to MSSA (Boyce Effect). Infections/colonizations by susceptible and resistant bacteria can coexist or occur sequentially without specific solution of continuity. The assumption made could therefore constitute a limitation to be acknowledged.

- It might be useful to describe in words in the body of the paper what is meant by "organized, intermediate or overwhelmed" responses. In the discussion, some considerations could also be introduced on how the availability of staff could influence the type of response. For example, the supplementary materials show how the "patient:HCW ratio" can influence compliance to the hand hygiene practice (Figure S1).

- Some definitions are not immediately clear. For example, it could be useful to describe in words the meaning for the indicator "average delay between compliant handwashing events".

Reviewer #4: I am not legitimate to evaluate the mathematical model elaborated by the authors and the theoretical results it produced.

I put "minor revisions" because the PLOSMed website does not allow more precise choice such as "pending evaluation by other appropriate experts".

In addition, there is no window to fill with "reviewer comments to editor blind to authors", in which i may write that the assumptions underlying the model are disputable, as there is no "Simultaneous transmission dynamics of SARS-CoV-2 () and a commensal bacterium ()", and the transmission of drug-resistant bacteria occur mainly within the healthcare facilities, not outside of them. 

I have several remarks regarding the assumptions underlying this model, and regarding the interpretation of these results. 

Introduction:

- line 83: there has not been such thing as an "antibiotic prophylaxis" in patients with Covid-19

- line 85-86: this hypothesis relies on the fact that drug-resistant bacteria are more transmitted out of the institutions (hospital, facility for the Elderly, etc), through daylife person-to-person contact (at home, at work, during leisures, during other interactions). Are the authors certain that it is the case in Europe? Such tranmission probably occur more frequently in healthcare institutions.

Methods

- lines 119-120: "Simultaneous transmission dynamics of SARS-CoV-2 () and a commensal bacterium ()": SARS-CoV-2 transmission occurs mainly by aerosol and droplets, meanwhile most of bacteria concerned by antibiotic resistance are transmitted by hand contact. Their transmission is not "simultaneous".

End of the abstract, and line 459 of the discussion, and line 555 in the conclusion: the authors cannot say that "Our model demonstrates how..." or "These findings thus highlight that...": being only in silico modelisation, it cannot "demonstrate" such things, but "suggest" or "bring arguments".

Discussion:

- lines 492-3: the author speak of the impact of the pandemic "leading to reduced whole genome sequencing of bacterial isolates" as if it was something done daily by any hospital laboratory in France, which is not appropriate.

- line 505-6: it is not possible to cite azithromycin and hydroxychloroquin as "treatment for COVID-19 and/or prophylaxis for bacterial coinfection" as they were not efficient, not recommended (appart from 2 weeks for HCQ in France in March 2020), and never demonstrated any effect on viral nor bacterial events in a Covid-19 context. It is not possible in the end of 2022 not to mention this when mentioning azithromycin and hydroxychloroquin.

[LINK]

---

## [Decision Letter · Decision Letter 2]

28 Apr 2023

Dear Dr. Smith,

Thank you very much for re-submitting your manuscript "Collateral impacts of pandemic COVID-19 drive the nosocomial spread of antibiotic resistance: a modelling study" (PMEDICINE-D-22-03098R2) for review by PLOS Medicine.

I have discussed the paper with my colleagues and the academic editor and it was also seen again by 2 reviewers. I am pleased to say that provided the remaining editorial and production issues are dealt with we are planning to accept the paper for publication in the journal.

[LINK]

We look forward to receiving the revised manuscript by May 05 2023 11:59PM.   

Sincerely,

Philippa Dodd, MBBS MRCP PhD

PLOS Medicine

plosmedicine.org

Requests from Editors:

GENERAL

Thank you for your detailed and considered responses to previous editor and reviewer requests and comments. Please see below for further minor revisions.

INTRODUCTION

Please check for formatting errors pertaining to in-text reference callouts, for example line 122 “(e.g. Streptococcus pneumoniae).[3]” punctuation should follow parentheses as follows, “(e.g. Streptococcus pneumoniae) [3].” Similarly, line 137. Please check and amend throughout.

Please indicate whether your study is novel and how you determined that. 

If there has been a systematic review of the evidence related to your study (or you have conducted one), please refer to and reference that review and indicate whether it supports the need for your study.

FINANCIAL DISCLOSURE

Line 666 – please remove this statement from the main manuscript and include only in the relevant part of manuscript submission form when you re-submit your manuscript. It will be compiled as metadata.

SOCIAL MEDIA

To help us extend the reach of your research, if not already done so, please provide any Twitter handle(s) that would be appropriate to tag, including your own, your coauthors’, your institution, funder, or lab. 

Please detail these in the relevant part of the manuscript submission form when you re-submit your manuscript.

Comments from Reviewers:

Reviewer #1: The authors have addressed all my points.

Michael Dewey

Reviewer #3: The revised version of the paper can be accepted for publication.

[LINK]

---

## [Editor Report · Decision Letter 3]

9 May 2023

Dear Dr Smith, 

On behalf of my colleagues and the Academic Editor, Dr. Ramanan Laxminarayan, I am pleased to inform you that we have agreed to publish your manuscript "Collateral impacts of pandemic COVID-19 drive the nosocomial spread of antibiotic resistance: a modelling study" (PMEDICINE-D-22-03098R3) in PLOS Medicine.

PRESS

Best wishes,

Pippa 

Philippa Dodd, MBBS MRCP PhD 

PLOS Medicine